# Antifungal Activities of Essential Oils in Vapor Phase against *Botrytis cinerea* and Their Potential to Control Postharvest Strawberry Gray Mold

**DOI:** 10.3390/foods11192945

**Published:** 2022-09-21

**Authors:** Dana Tančinová, Zuzana Mašková, Andrea Mendelová, Denisa Foltinová, Zuzana Barboráková, Juraj Medo

**Affiliations:** 1Institute of Biotechnology, Faculty of Biotechnology and Food Sciences, Slovak University of Agriculture in Nitra, Tr. A Hlinku 2, 949 76 Nitra, Slovakia; 2Institute of Food Sciences, Faculty of Biotechnology and Food Sciences, Slovak University of Agriculture in Nitra, Tr. A Hlinku 2, 949 76 Nitra, Slovakia; 3The State Veterinary and Food Administration of the Slovak Republic, Botanická 17, 842 13 Bratislava, Slovakia

**Keywords:** essential oils, strawberries, *Botrytis*

## Abstract

Essential oils (EOs) from aromatic plants seem to have the potential to control several fungal pathogens and food contaminants. *Botrytis cinerea* is the main strawberry fruit contaminant causing high losses during storage. Here, thirteen EOs applied in the vapor phase were evaluated for their potential to inhibit the growth of three different strains of *B. cinerea* isolated from strawberry fruits. Eight EOs (lemongrass, litsea, lavender, peppermint, mint, petitgrain, sage, and thyme) were able to completely inhibit the growth of *B. cinerea* for 7 days when applied at a concentration of 625 μL·L^−1^. Four EOs with the lowest minimal inhibition concentrations (thyme, peppermint, lemongrass, and litsea) have been tested on strawberry fruits intentionally inoculated by *B. cinerea*. All four EOs showed high inhibition at a concentration of 250 or 500 μL·L^−1^, but only peppermint EO was able to completely inhibit *B. cinerea* lesion development at a concentration of 125 μL·L^−1^. The sensory evaluation of strawberries treated by EOs at a concentration 125 μL·L^−1^ resulted in a statistically significant decrease in taste, aftertaste, aroma, and overall quality. Lemongrass and litsea EOs scored better than thyme and peppermint ones, thus forming two viable methods for *B. cinerea* suppression and the extension of packed strawberries’ shelf life.

## 1. Introduction

Strawberries are one of the most cultivated fruits in the world, and their cultivation is constantly growing [1,2,3]. Strawberry fruits have a very short shelf life, and significant post-harvest losses occur in the fresh-product supply chain. The quality of fruit is rapidly reduced over time, resulting in up to 40% losses [4]. The degradation of berries is mainly caused by fungal pathogens [2,5]. The most common cause of decay of strawberries is gray mold, also called Botrytis rot as it is caused by *Botrytis cinerea*. This species grows rapidly and destroys strawberry fruit within a few days. The disease can begin pre harvest, remain as a hidden infection, or start after harvest [6,7,8,9]. Up to 77% of strawberry fruit samples were contaminated by this fungal species [10].

Post-harvest rot control is traditionally achieved using chemical fungicides [11]. However, there is an increasing tendency among consumers to refuse chemical treatments. Furthermore, natural substances for the extension of the shelf life of food are preferred today. Essential oils (EOs) are considered suitable substitutes for chemical food preservatives [8,12,13,14,15,16,17,18]. They appear to be effective against various species of microorganisms that are resistant to other preservatives [19]. EOs and their components are also important because of their availability, their range of biological activities, and their low cost [20].

Depending on the source plant species, EOs contain various active ingredients such as phenols (thymol, carvacrol), alcohols (linalool, menthol), aldehydes (citral, cinnamaldehyde), and others. These compounds can efficiently stop the growth of microorganisms through the inhibition of microbial metabolism and gene expression, cell wall degradation, the blockage of DNA repair, and other mechanisms [21]. In addition, some EOs showed antioxidant properties that may help to extend the shelf life of fruits [22].

The effectiveness of the EO treatment in the management of fungal pathogens after harvest depends on the cultivar of the fruit, the composition and the concentration of the applied EO, the duration of storage, and the method of application [23].

There is growing evidence that the application of EOs in the vapor phase is an effective antimicrobial treatment and has advantages over liquid-phase EOs applications, such as increased activity, use at lower concentrations, and the ability to be used in a variety of environments [24]. Due to these advantages, many studies on the possible use of EOs in the vapor phase during fruit storage have been carried out in past years [25,26,27,28]. Nevertheless, there is no general uniformity regarding the effectiveness of EOs against various microorganisms, and therefore it is necessary to experimentally identify the activity of each EO vapor with regard to particular microbial species [29].

In recent years, the antifungal activity of EOs against *B. cinerea* has been examined by several studies [9,13,30,31,32,33,34,35,36,37], and some of EOs have shown promising properties regarding *B. cinerea* inhibition on various fruits or vegetables. Additionally some EOs like tea tree [38], thyme [39], lemon, cinnamon [40], oregano, and ziziphora [41] were evaluated for strawberry fruit preservation. Some studies have yielded results that do not correspond to other studies, although the same EOs are used. Thus, there is still a challenge to find candidate EOs for strawberry preservation due to differences in treatments, doses, and the origin of fungal isolates. Excellent EOs antifungal activity often does not correlate with the potential of real use. EOs have a strong flavor and scent that may not be accepted by consumers [42]. Thus, only some of the EOs are suitable for the preservation of particular foods.

The aim of the presented research was to measure the ability of selected EOs to inhibit the growth of *B. cinerea* strains, and to select the most effective EOs and evaluate their potential to control *B. cinerea* development directly on strawberries. Finally, the impact of these EOs on the sensory properties of strawberries was considered. We hypothesized that some of the EOs are able to effectively suppress *B. cinerea* and provide an extension of storage without negative effects on strawberry sensory traits.

## 2. Materials and Methods

### 2.1. Fungal Strains

Strains of *Botrytis cinerea* used in assays were isolated from moldy strawberries in 2020. Strain *B. cinerea* KMi-284 was isolated from packed strawberries obtained from supermarket (fruit origin–Spain). Strains KMi-507 and KMi-508 were isolated from strawberries obtained from local fresh market in Nitra (Western Slovakia) and Banská Bystrica (Middle Slovakia), respectively. Strains were identified using a polyphasic system involving micro- and macro-morphological traits and molecular methods. The internal transcribed spacer DNA sequences of the strains were deposited in GenBank under accession numbers ON318873-5. The strains of *B. cinerea* were deposited in the Collection of Microorganisms of the Institute of Biotechnology, Department of Microbiology, Faculty of Biotechnology and Food Science, SUA in Nitra, Slovakia.

For growth-inhibition assays, the strains were grown on potato-dextrose agar (PDA; HIMEDIA India) at 22 ± 1 °C for 7 days. Spores were collected by rinsing the colony with physiological saline solution supplemented with Tween 80 (0.5%). Conidial suspension with a concentration of 10^4^ spores/mL was prepared for each fungal strain. The EVE^TM^ automatic cell counter (NanoEnTek, Seoul, Korea) was used to determine the number of spores.

### 2.2. EOs

Thirteen commercially available EOs from seven plant families were used in the assay. According to the information provided by the producers, the EOs were obtained by hydro-distillation. The semi-quantitative composition of the EO samples was determined by gas chromatography coupled with mass spectrometry (GC-MS) using an Agilent 7890B oven coupled with Agilent 5977A mass detector (Agilent Technologies Inc., Palo Alto, CA, USA) and CombiPal autosampler 120 (CTC Analytics AG, Zwingen, Switzerland). The methodology for the determination of EOs components was detailed in our previous study [43]. Details about the composition of the EOs are listed in Table 1.

### 2.3. Fungal Growth Inhibition Assays

We used a multi-step selection strategy where only promising EOs were used in further steps for the complex evaluation of the inhibitory potential of the EOs. The first step involved testing the inhibitory effects in vitro using high doses of EOs, and the second step comprised in vitro testing of several concentrations of EOs and the determination of inhibitory concentrations. The in vivo assay with the strawberries infected by *B. cinerea* represented the third step. The final step was sensory analysis to evaluate the potential of EO use in strawberry packaging.

#### 2.3.1. In Vitro Testing of Inhibitory Effect

The vapor-phase diffusion method was used to determine the inhibitory effect of EOs on *B. cinerea* growth. Strains were cultivated on PDA for 7 days in 22 ± 1 °C. Petri dishes with 9 cm diameter were filled by 15 mL of PDA medium. Five microliters of spore suspension prepared as mentioned in 2.1 were inoculated in the center of the media. A small piece of Whatman No.1 filter paper (1 cm × 1 cm) was placed in the center of the Petri dish cover and infused by 50 µL of concentrated EO. The Petri dish was sealed by parafilm M and cultivated in an upside-down position. The evaporated EO provided a concentration of 625 µL of EO in one liter of air. The experiment was carried out in triplicates. Fifty microliters of dimethylsulfoxide (DMSO) were used instead of EOs in a control treatment. The growth of colonies was observed on the 3rd, 4th, and 7th day of cultivation. The diameter of the colonies was evaluated using a digital caliper. The antifungal activity of the EOs was expressed by relative inhibition calculated using Equation (1), where RI is the relative inhibition in %, c is the diameter of the colony in the control, and t is the diameter of the colony treated by EO.

Equation (1):RI = [(c − t)/c] × 100(1)

#### 2.3.2. Determination of Inhibitory Concentrations

Minimal inhibitory concentrations (MICs) were estimated only for EOs that showed a 100% inhibitory effect in the previous step with 625 μL·L^−1^ concentration. For this purpose, EOs were diluted in DMSO to a concentration that provided 500 μL·L^−1^ in vapor phase when the oil was applied to the paper in amounts of 50 μL. This concentration was serially diluted in DMSO to obtain concentrations of 250, 125, 62.5, 31.25, and 15.625 μL·L^−1^ in the vapor phase. Six replications were conducted for each dose. The presence of fungal growth was evaluated on the 3rd, 4th, and 7th day of cultivation and probit analysis was used for the estimation of inhibitory doses when 50% (IC50) or 90% (IC90) of the colonies were not able to grow.

#### 2.3.3. In Vivo Evaluation of Antifungal Activity of EOs on Strawberries

The strawberries were purchased directly from the local Slovak grower. The experiment was established on the day of the collection and purchase of fruits. Fruits were selected to have the same weight without any signs of infection or mechanical damage. The strawberries were treated with a freshly prepared 1% sodium hypochlorite solution to minimize microbial contamination from the field. Further, the fruits were rinsed with sterile water and dried at room temperature. Cleaned strawberries were placed in small clear plastic containers. Four 1 mm wounds were made on each strawberry with a sterile tip in the equatorial plane. Then, 5 µL of *B. cinerea* spore suspension (10^4^ spores in 1 mL) was added to the wound site with a micropipette. Containers with strawberries were placed in sealable glass jars with a volume of 500 mL. Whatman No. 1 filter papers (diameter 50 mm) were placed in the cup closures and EOs were applied. Thyme, litsea, peppermint, and lemongrass were used in the assay. EOs were prepared in 3 concentrations (100%, 50% and 25%) using DMSO as solvent, and 250 μL of solution was applied to filter paper providing EO vapor concentrations of 500, 250, and 125 μL·L^−1^. DMSO (250 µL) was applied instead of EO as a control. All variants had three replicates. The glasses were covered with foil to prevent access to light and stored at room temperature (21 ± 1 °C). The growth of *B. cinerea* was monitored on the 3rd, 4th, 5th, 6th, and 7th day after the inoculation. The number of lesions (0–12) developed in 12 inoculating points (4 points in each replication), was scored in each treatment.

### 2.4. Sensory Analysis

The strawberries were placed in sealable transparent glass cups with a volume 500 mL. Strawberries (5 pieces) were selected to have the same weight. The EOs at a concentration 125 µL·dm^−3^ were applied to Whatman No. 1 filter papers in a cup cap. The glasses were stored in a refrigerator at 3 ± 1 °C for 5 days. All variants had three replicates.

Sensory quality was rated on a 9-point scale (1–2 represented extreme dislike; 3–5 fair; 6–8 good; and 9 excellent) for appearance, aroma, taste, aftertaste, and overall acceptability. At the beginning of the experiment, 5 panelists were trained to evaluate the relevant characteristics of the fruit. Sensory evaluation was performed in a sensory laboratory equipped with separate sensor boxes.

### 2.5. Statistical Analysis

Data from fungal inhibition analysis and sensory analysis were evaluated using one-way variance analysis (ANOVA) followed by a post hoc Tukey HSD test. Inhibitory concentrations of IC50 and IC90 were estimated using probit analysis [44]. All statistical analyses were carried out in an R environment [45].

## 3. Results

### 3.1. Evaluation of EOs Inhibitory Properties

Among thirteen evaluated EOs, nine oils demonstrated absolute inhibition of *B. cinerea* growth in the first step of our multi-level evaluation when 625 μL·L^−1^ concentration of EO vapor was tested (Table 2). The antifungal activity of EOs expressed as the relative inhibition of fungal growth is summarized in Appendix A. EOs from lemongrass, litsea, lavender, peppermint, mint, petitgrain, sage, and thyme inhibited the growth of all strains during the whole period of 7 days. After eucalyptus EO treatment, the growth of a single strain, i.e., KMi-507, was detected on the 4th and 7th day, while other strains remained completely inhibited. Grapefruit EO caused a delay in the growth of fungal colonies as there was no measurable growth on the second day. However, colony growth was detected on the third day, and finally on 7th day fungus overgrew the whole Petri plate, similarly to the colonies in the untreated control. Ginger EO acted similarly, but the inhibitory level was lower. Jasmine EO did not significantly slow down growth, and it even stimulated the growth of strain KMi-284 on the 2nd day. The last four mentioned EOs have been removed from further testing.

### 3.2. Inhibitory Concentrations of EOs

The lowest MICs were observed for thyme, litsea, and peppermint EOs. These EOs completely inhibited the growth of any strain of *B. cinerea* on the 7th or 14th days at a concentration of 250 μL·L^−1^. The EOs concentration of 500 μL·L^−1^ inhibited growth when lemongrass and lavender EOs were used. Cardamom, mint, petitgrain, and sage EOs inhibited growth completely only when used in the highest dose (625 μL·L^−1^).

We estimated concentrations that inhibited B. cinerea growth in 50% or 90% of cases (IC50 and IC90) by probit analysis for each fungal strain and EO (Figure 1). The values of IC50 varied greatly among EOs, as well as among strains, which possess different reactions of particular strain to EO treatment. For example, the strain KMi-507 was the most sensitive to the presence of thyme EO (IC90 = 94.61 μL·L^−1^), but it was the most resistant to litsea EO treatment (IC90 = 128.65 μL·L^−1^) on the 7th day. IC90 values were not very different from IC50, suggesting that the threshold for growth inhibition is relatively sharp and the cessation of growth occurs relatively shortly after the specific concentration of EO in the vapor phase is reached. The lowest values of IC50 were detected for thyme EO, followed by litsea, mint, and lemongrass. They were substantially more effective than lavender, petitgrain, and sage EOs.

### 3.3. Inhibition of B. cinerea on Strawberries

The real ability of previously selected EOs to suppress *B. cinerea* during the storage of strawberries was tested in three concentrations 125, 250, and 500 μL·L^−1^ (Table 3). The development of *B. cinerea* lesions was observed in all 12 inoculation points on control strawberry fruits. On the other hand, lesions were not developed in any 500 μL·L^−1^ treatment. The EOs applied at a concentration of 250 μL·L^−1^ inhibited *B. cinerea* development in all cases except lemongrass EO against the KMi-508 strain and litsea EO against strains KMi-284 and KMi-508. The lowest tested concentration also inhibited lesion development despite the fact its action was not sufficient in case of litsea, lemongrass, or thyme. Peppermint EO at a concentration of 125 μL·L^−1^ did not allow for the development of the *B. cinerea* lesion at all.

### 3.4. Sensory Analysis of Strawberries Treated by EOs

Samples of strawberries treated with the 125 μL·L^−1^ concentrations of EOs were also evaluated from a sensory point of view after 5 day storage. Higher concentrations of EOs, e.g., 250 and 500 μL·L^−1,^ were evaluated as unacceptable in the preliminary assay.

In terms of statistical significance (*p* < 0.05), the best sensory quality in terms of taste, aftertaste, and overall acceptability was achieved by the control sample (Appendix A). The treatment of samples with EOs did not show a statistically significant change in the appearance of the samples. All of the samples had a fresh appearance, the fruits were shiny, and the stem of the fruit was green and unadulterated. Sample treatment by EOs resulted in statistically significant differences in taste, aftertaste, aroma, and overall quality (Figure 2). In the aroma trait, the best results were achieved by samples treated with lemongrass EO. It was characterized by a dominant strawberry aroma. In the samples treated with peppermint and thyme EOs, the scores dropped to 5.73 and the evaluators described the aroma of these samples as pleasant, but the typical strawberry aroma was overpowered by the EO.

There was a decrease in taste and aftertaste scores by 1.86–3.60 points. The highest scores were achieved by samples treated with lemongrass, litsea, and peppermint EOs. The taste of the samples was considered to be good. The taste and aftertaste were, statistically speaking, significantly worse in the samples treated with thyme EO. The taste of these samples was acceptable, but the flavor of the used EO dominated over the natural strawberry flavor.

## 4. Discussion

In this study, we assessed the effect of EOs from 13 different plant species against *B. cinerea*. The quality and composition of used EOs are crucial in evaluating their effects, because the quality can significantly affect the results. Many studies [46,47,48,49] confirmed variation in the composition of EOs depending on the growing season, the nature of plant parts, and different stages of plant growth and climatic conditions. According to GC-MS analysis, used commercially available EOs had compositions in line with other published studies, and the main components were in the typical range for certain types of EOs.

According to the results, EOs were divided into three groups. The first group consisted of EOs that had a weak inhibitory effect on the strains of *B. cinerea.* This group included grapefruit, jasmine, and ginger EOs. The efficiency of grapefruit and ginger EOs declined rapidly. The least effective EO was the jasmine one, which even stimulated the growth of fungal colonies. However, the inhibitory effect of these EOs on microscopic fungi was reported in the research of other authors. According to Viuda-Martos et al. [50], grapefruit EO was the best in terms of the growth reduction in *Penicillium chrysogenum* and *Penicillium verrucosum* when several citrus EOs were tested. Kujur et al. [51] showed significant protection of maize seeds against fungal infection after treatment by nano-encapsulated jasmine EO. Jasmine EO also showed the antibiofilm activity of *Candida albicans* [52]. According to Nerilo et al. [53], only a low concentration of ginger EO was needed to curb the production of aflatoxin B1 by *Aspergillus flavus*. 

The second group consisted of effective EOs that inhibited the growth of the strains tested: eucalyptus and cardamom EOs. Eucalyptus EO 100% inhibited the growth of two of the three *B. cinerea* strains. Davari et Ezazi [54] reported an inhibitory effect of eucalyptus EO on *B. cinerea* from 0 to 84.88, depending on the concentration used. They rated this EO as moderately or poorly effective. Similarly, other authors report that eucalyptus EO is effective against fungi but only in higher concentrations [55].

In our essay, *Botrytis cinerea* growth in the presence of cardamom EO was recorded in only one strain (KMi-284) and only in the last measurement (day 7). The effectiveness of inhibition at this time point was high, 83.33%. Antibacterial action and the suppression of biofilm formation was previously recorded for cardamom extracts [56]. Traditional medicine has been using the cardamom EO for a long time, and it is a promising compound in the fight against acute campylobacteriosis [57,58]. Cardamom EO effectively inhibited *A. flavus* growth in peanuts and reduced aflatoxin production [59].

The third group of EOs in our study comprised thyme, litsea, peppermint, lemongrass, lavender, petitgrain, mint, and sage EOs. These completely inhibited the growth of *B. cinerea* strains at a concentration 625 μL·L^−1^. Out of them, thyme, litsea, peppermint, and lemongrass showed the best antifungal properties in further testing.

In correspondence with our results, a significant antifungal effect on *Fusarium* species (*F. avenaceum*, *F. culmorum*, *F. graminearum*, and *F. oxysporum*) has been reported after treatment by thyme, litsea, lemongrass, and verbena EOs. Their effect was comparable to a synthetic pesticide Funaben T [60]. Litsea EO applied by the agar dilution method at a concentration of 1.0% resulted in the complete inhibition of *B. cinerea* growth [36]. Amiri et al. [35] showed that the use of the peppermint EO and savory EO applied in the vapor phase was more effective at inhibiting the growth of *B. cinerea* than the liquid application. These EOs had a significant impact on the growth of B. cinerea at reasonably low concentrations. Moreover, the application of EOs in the vapor phase is easily adaptable for the food industry. 

In the experiment carried out by Reang et al. [37], thyme was the best evaluated in terms of the growth inhibition of *B. cinerea* among five EOs (clove, thyme, lavender, lemongrass, and peppermint). All of these EOs inhibited the growth of B. cinerea when tested by the agar dilution method, but the efficiency of inhibition varied between the EOs and the concentration used. At a concentration of 1.5%, thyme EO inhibited the growth of mycelia by 50%, clove 44.65%, lemongrass 40.89%, lavender 40.35%, and peppermint 38.39%. Thyme EO significantly (by 64%) reduced the colonization of detached tomato leaves by *B. cinerea* when applied by foliar spraying. [61].

The best evaluated EOs have different main components, and their mode action in fungal inhibition is not the same. For example, thyme EO affected the growth of fungus *Mycosphaerella graminicola* through the regulation of the expression of genes involved in cell development and detoxification [62]. Citral affected the cell membrane [63], but its mechanism of action does not involve the cell wall or ergosterol [64]. Detailed knowledge of the mode of action is still lacking for most of the EOs compounds. Moreover, the interaction of particular compounds within an EO plays an important role [65].

Ansarifar et Moradinezhad [39] showed promising ways to preserve strawberries using thyme EO encapsulated in zein nanofibers. This packaging led to a decrease in bacterial and fungal development, while acidity, total phenol content, and antioxidant activity were maintained. Despite the authors examining the appearance of fruit, they did not report changes in flavor or taste.

As mentioned earlier, the strong organoleptic properties of EOs are a complication for their use in the preservation of food. For this reason, we also evaluated the sensory properties of strawberries after the application of EO. In our essay, the aroma, flavor, and aftertaste of the strawberries were significantly overpowered by peppermint EO. In terms of overall acceptability, strawberries samples treated with lemongrass EO received the highest score. The fruits retained typical strawberry properties, and the EO did not interfere with the character of the aroma and flavor. Of all the EOs tested, the panelist describes lemongrass EO as the most compatible with strawberries in sensory traits. Citral, which is the main component of lemongrass EO, was recently positively evaluated for *Rhizopus oryzae* control on table grapes [25]. After application at a dose of 0.0125 µL·cm^−3^, the panelists still recognized the odor of citral in grape samples. However, the authors of the study declared the odor diminishing 30 min after the package opening.

Yanzhen et al. [38] treated strawberries after harvesting with tea tree EO at a concentration of 0.3–0.9 g·L^−1^ air; then, they left the fruit for 3 h in the EO environment. After 3 days of storage, all treatments significantly (*p* < 0.05) maintained higher sensory scores than those of control groups, which indicated that all these treatments help to maintain the color, aroma quality, and overall acceptability of strawberry fruit. The treatment of strawberries with *Solidago canadensis* EO in the vapor phase effectively suppressed the growth of *B. cinerea* and preserved the postharvest quality. Additionally, the sensory acceptance of the strawberries was higher than the control in the 2nd, 3rd, and 4th day [66]. However, a sensory analysis of the same treatment can result in contrary results, e.g., Shehata et al. [67] described the increase of sensory traits after strawberry treatment by lemon EO, while Perdones et al. [34] evaluated the effect of the same oil in the combination with chitosan negatively. Gol et al. [5] treated strawberries only with chitosan after harvesting. After 8 and 12 days, they found significantly better sensory quality in the coated samples than in the untreated samples.

It is important to note that our strawberries in sensory tests were not decayed and did not show any microbial damage in control and treated samples at the time of the test (on the 5th day). Authors of some studies [66,67] used the conditions (long storage or higher temperature) in which fruits in the untreated control decayed, while samples treated by EOs scored better due to EOs antimicrobial activity. Although a test in these conditions reflects reality, it can also hide the negative sensory effects of EOs. Minor changes of aroma or flavor caused by EOs are significantly lower than changes due to microbial activity [67]. Such tests partially lack relevance because consumers are not supposed to buy or consume decayed fruit. The evaluation should be carried out in non-decayed fruits to reveal the true impact of EOs on sensory traits.

Despite the fact thyme EO showed the best inhibitory action in our essay, we cannot recommend its use due to the changed sensory values. Small fruits are exceptionally challenging in this because consumers have high demand for the natural and original sensory properties of the fruit. Based on a complex view of all of the tested EOs, considering their ability to preserve strawberries along with their drawbacks in terms of potential customer acceptance, lemongrass EO and (to some extent) litsea EO are good candidates for use in the food industry. In recent years, encapsulation, nanoparticles, and substances capable of creating an edible coating with high preservation ability, such as chitosan, have gained attention [68,69,70,71,72]. The combination of the high effective EOs with these technologies in active packaging can improve the shelf life of many foods, including fruits such as strawberries. As EOs are natural substances with the potential ability to extend the shelf-life of fruits, they can be a healthier choice for consumers than the use of inorganic substances in active packaging [73].

## 5. Conclusions

Four of thirteen evaluated EOs showed promising levels of *B. cinerea* growth inhibition and decreased lesion development in packed strawberries. However, EOs more or less changed the sensory quality of strawberries. Lemongrass and litsea EOs seem to be acceptable for consumers when applied at 125 μL·L^−1^ concentration. The effect may be strengthened by storing them at lower temperatures. However, the selection of EOs with good inhibition properties and without negative sensory effects is desirable.

## Figures and Tables

**Figure 1 foods-11-02945-f001:**
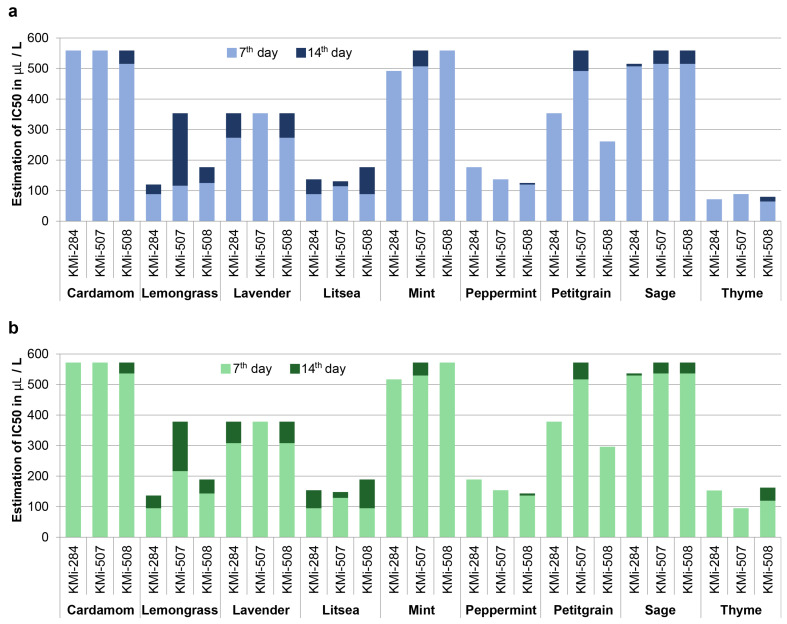
Graph of estimated concentration of essential oils that inhibited growth of *B. cinerea* on 50% (**a**) and 90% (**b**) inoculated plates.

**Figure 2 foods-11-02945-f002:**
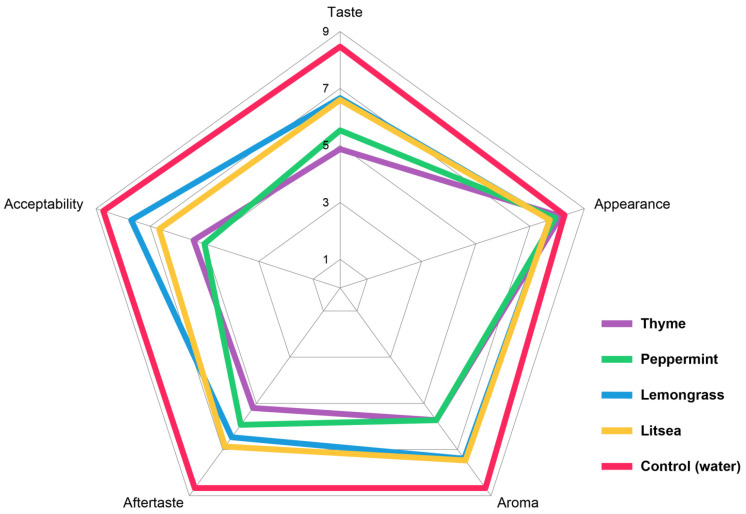
Radar plot of the sensory evaluation of strawberries treated by essential oils in vapor phase.

**Table 1 foods-11-02945-t001:** Main compounds of essential oils (EOs) used in *Botrytis cinerea* inhibition assay determined by gas chromatography coupled with mass spectrometry (GC-MS). Only compounds representing more than 5% of particular EO are listed.

Essential Oil	Plant	Compound	Occurrence in %
Cardamom	*Elettaria cardamomum* L.	α-Terpineol acetate	43.7
		1,8-Cineole	33.1
Eucalyptus	*Eucalyptus globulus* L.	1,8-Cineole	79.30
		(R)-(+)-Limonene	6.90
		p-Cymene	6.30
Ginger	*Zingiber officinale* Roscoe	(−)-Zingiberene	34.7
		α-Curcumene	13.2
		Cadinene, (+)-	13.1
		Farnesene	12.5
		β-Myrcene	7.7
Grapefruit	*Citrus × paradisi* Macfadyen	1,8-Cineole	92.2
Jasmine	*Jasminum officinale* L.	(−)-Borneol	37.7
		(R)-(+)-Limonene	19.2
		Benzyl benzoate	10.9
Lavender	*Lavandula angustifolia* Mill.	(−)-Linalool	35.5
		(−)-Bornyl acetate	35.1
Lemongrass	*Cymbopogon flexuosus* Nees ex. Steud	α-Citral	35.2
		β-Myrcene	28.3
		β-Citral	28.3
Litsea	*Litsea cubeba* (Lour.) Pers	α-Citral	38
		β-Citral	31.4
		(R)-(+)-Limonene	14.6
Mint	*Mentha aquatica* L. var. *citrata* (Her.)	Geraniol	42.1
		(−)-Linalool	37.2
		Geranyl acetate	5.2
Peppermint	*Mentha × piperita* L.	Menth-1-en-4-ol	42.5
		(±)-Citronellal	23.1
		(−)-Borneol	8.5
		1,8-Cineole	7.2
		(+)-α-pinene	6.0
		2-Undecanone	5.8
Petitgrain	*Citrus × aurantium* L.	(R)-(+)-Limonene	31.9
		(−)-β-Pinene	17.5
		α-Citral	12.2
		β-Citral	8.2
		Geraniol	6.9
		Geranyl acetate	6.7
Sage	*Salvia officinalis* L.	α-Thujone	23.0
		(−)-Isopulegol	20.1
		1,8-Cineole	11.0
		(−)-Alloaromadendrene	7.0
		Camphene	6.0
		fenchyl alcohol	6.0
		(+)-α-pinene	5.1
Thyme	*Thymus vulgaris* L.	(+)-Menthofuran	51.5
		p-Cymene	16.5
		β-Caryophyllene	5.1

**Table 2 foods-11-02945-t002:** Average diameter of *Botrytis cinerea* colonies on potato dextrose agar (22 ± 1 °C) after treatment by essential oils in vapor phase (625 μL·L^−1^).

Strain	KMi284				KMi507				KMi508				
Day	2nd	3rd	4th	7th	2nd	3rd	4th	7th	2nd	3rd	4th	7th	
Control	23.2 ^b^	56.0 ^d^	90.0 ^c^	90.0 ^c^	36.2 ^d^	56.8 ^e^	90.0 ^d^	90.0 ^c^	30.0 ^c^	65.5 ^e^	90.0 ^d^	90.0 ^b^	30.0 ^c^
Cardamom	0.0 ^a^	0.0 ^a^	0.0 ^a^	10.5 ^b^	0.0 ^a^	0.0 ^a^	0.0 ^a^	0.0 ^a^	0.0 ^a^	0.0 ^a^	0.0 ^a^	0.0 ^a^	0.0 ^a^
Eucalyptus	0.0 ^a^	0.0 ^a^	0.0 ^a^	0.0 ^a^	0.0 ^a^	0.0 ^a^	11.7 ^b^	58.2 ^b^	0.0 ^a^*	0.0 ^a^	0.0 ^a^	0.0 ^a^	0.0 ^a^*
Ginger	0.0 ^a^	10.3 ^b^	49.3 ^b^	90.0 ^c^	19.0 ^b^	39.3 ^c^	90.0 ^d^	90.0 ^c^	7.8 ^b^	37.7 ^c^	62.8 ^c^	90.0 ^b^	7.8 ^b^
Grapefruit	0.0 ^a^	13.5 ^c^	49.2 ^b^	90.0 ^c^	0.0 ^a^	30.0 ^b^	51.0 ^c^	90.0 ^c^	0.0 ^a^	26.3 ^b^	52.8 ^b^	90.0 ^b^	0.0 ^a^
Jasmine	27.7 ^c^	56.3 ^d^	90.0 ^c^	90.0 ^c^	33.2 ^c^	50.3 ^d^	90.0 ^d^	90.0 ^c^	30.0 ^c^	56.3 ^d^	90.0 ^d^	90.0 ^b^	30.0 ^c^
Lavender	0.0 ^a^	0.0 ^a^	0.0 ^a^	0.0 ^a^	0.0 ^a^	0.0 ^a^	0.0 ^a^	0.0 ^a^	0.0 ^a^	0.0 ^a^	0.0 ^a^	0.0 ^a^	0.0 ^a^
Lemongrass	0.0 ^a^	0.0 ^a^	0.0 ^a^	0.0 ^a^	0.0 ^a^	0.0 ^a^	0.0 ^a^	0.0 ^a^	0.0 ^a^	0.0 ^a^	0.0 ^a^	0.0 ^a^	0.0 ^a^
Litsea	0.0 ^a^	0.0 ^a^	0.0 ^a^	0.0 ^a^	0.0 ^a^	0.0 ^a^	0.0 ^a^	0.0 ^a^	0.0 ^a^	0.0 ^a^	0.0 ^a^	0.0 ^a^	0.0 ^a^
Mint	0.0 ^a^	0.0 ^a^	0.0 ^a^	0.0 ^a^	0.0 ^a^	0.0 ^a^	0.0 ^a^	0.0 ^a^	0.0 ^a^	0.0 ^a^	0.0 ^a^	0.0 ^a^	0.0 ^a^
Peppermint	0.0 ^a^	0.0 ^a^	0.0 ^a^	0.0 ^a^	0.0 ^a^	0.0 ^a^	0.0 ^a^	0.0 ^a^	0.0 ^a^	0.0 ^a^	0.0 ^a^	0.0 ^a^	0.0 ^a^
Petitgrain	0.0 ^a^	0.0 ^a^	0.0 ^a^	0.0 ^a^	0.0 ^a^	0.0 ^a^	0.0 ^a^	0.0 ^a^	0.0 ^a^	0.0 ^a^	0.0 ^a^	0.0 ^a^	0.0 ^a^
Sage	0.0 ^a^	0.0 ^a^	0.0 ^a^	0.0 ^a^	0.0 ^a^	0.0 ^a^	0.0 ^a^	0.0 ^a^	0.0 ^a^	0.0 ^a^	0.0 ^a^	0.0 ^a^	0.0 ^a^
Thyme	0.0 ^a^	0.0 ^a^	0.0 ^a^	0.0 ^a^	0.0 ^a^	0.0 ^a^	0.0 ^a^	0.0 ^a^	0.0 ^a^	0.0 ^a^	0.0 ^a^	0.0 ^a^	0.0 ^a^

* Values followed by the same letter (in single column) are not significantly different on α = 0.05 ANOVA, Tukey HSD post-hoc test.

**Table 3 foods-11-02945-t003:** Development of *B. cinerea* lesions on strawberries treated by the essential oils in vapor phase.

Isolate	Essential Oil	Dose (μL·L^−1^)	Day				
			3rd	4th	5th	6th	7th
KMi-284	Control (DMSO)	0	12/12 *	12/12	12/12	12/12	12/12
	Lemongrass	125	4/12	5/12	6/12	6/12	6/12
		250	0/12	0/12	0/12	0/12	0/12
		500	0/12	0/12	0/12	0/12	0/12
	Litsea	125	2/12	3/12	3/12	3/12	3/12
		250	1/12	1/12	1/12	1/12	1/12
		500	0/12	0/12	0/12	0/12	0/12
	Peppermint	125	0/12	0/12	0/12	0/12	0/12
		250	0/12	0/12	0/12	0/12	0/12
		500	0/12	0/12	0/12	0/12	0/12
	Thyme	125	0/12	0/12	1/12	3/12	3/12
		250	0/12	0/12	0/12	0/12	0/12
		500	0/12	0/12	0/12	0/12	0/12
KMi-507	Control (DMSO)	0	12/12	12/12	12/12	12/12	12/12
	Lemongrass	125	0/12	0/12	0/12	0/12	0/12
		250	0/12	0/12	0/12	0/12	0/12
		500	0/12	0/12	0/12	0/12	0/12
	Litsea	125	2/12	2/12	2/12	2/12	2/12
		250	0/12	0/12	0/12	0/12	0/12
		500	0/12	0/12	0/12	0/12	0/12
	Peppermint	125	0/12	0/12	0/12	0/12	0/12
		250	0/12	0/12	0/12	0/12	0/12
		500	0/12	0/12	0/12	0/12	0/12
	Thyme	125	1/12	1/12	1/12	1/12	1/12
		250	0/12	0/12	0/12	0/12	0/12
		500	0/12	0/12	0/12	0/12	0/12
KMi-508	Control (DMSO)	0	12/12	12/12	12/12	12/12	12/12
	Lemongrass	125	0/12	1/12	1/12	2/12	2/12
		250	1/12	1/12	1/12	1/12	1/12
		500	0/12	0/12	0/12	0/12	0/12
	Litsea	125	1/12	3/12	3/12	4/12	4/12
		250	0/12	1/12	1/12	1/12	1/12
		500	0/12	0/12	0/12	0/12	0/12
	Peppermint	125	0/12	0/12	0/12	0/12	0/12
		250	0/12	0/12	0/12	0/12	0/12
		500	0/12	0/12	0/12	0/12	0/12
	Thyme	125	2/12	1/12	1/12	1/12	1/12
		250	0/12	0/12	0/12	0/12	0/12
		500	0/12	0/12	0/12	0/12	0/12

* number of developed *B. cinerea* lesions on 12 inoculation points on strawberry fruits.

## Data Availability

All data are available on correspondence author upon request.

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
