# Peer review of "Antifungal Activities of Essential Oils in Vapor Phase against *Botrytis cinerea* and Their Potential to Control Postharvest Strawberry Gray Mold"

_foods, 2022, doi:10.3390/foods11192945_

Round 1
Reviewer 1 Report
The manuscript explored the Antifungal Activities of Essential Oils in Vapor Phase against Botrytis cinerea and Their Potential to Control Postharvest Strawberry Gray Mold. The manuscript is clear in ideas, however, the logic and grammar need to be modified. Besides, there are several issues with the manuscript that need to be revised.
Materials and methods:
(1) Line 92-93: Please rewrite “For each fungal strain, a conidial spore suspension of 104 spores in mL-1 was prepared.”
(2) Line 108-109: Should this be divided into two sentences? “----Eos, we-----”
(3) Line 123: “-----in L. Experiment---“ ??? Please explain it!
(4) Line 126-129: Please use the typesetting method of formula.
(5) Line 133: What’s the concentration of DMSO added? Will it affect the growth of fungi?
(6) Line 135: “-----providing 250; 125; 62.5; 31.25, and ---”. Please correct “;’ to “,”.
Results:
(7) Line 195: Correct “Minimal inhibition concentrations” to “MICs”, because the “Materials and Methods” part has already appeared! In addition, essential oils and EOs have the same problem. Please check the details of the full manuscript.
(8) Line 218: “control treatment”?? It should be control samples.
(9) Line 220: What is the “The concentration of 250 µ l.L-1” mean? The subject of the sentence is EO or EO’s concentration, not concentration!
(10) Line 232: (p<0.05) should be (p < 0.05). P should be changed to italics.
Discussion:
(11) Line 296: Please rewrite this sentence.
(12) Line 331-333: “If we compared treated and untreated samples and allowed them to decay the non-treated fruits scores worse than EO treated due they damage.” Here the language is confused. Moreover, in the whole manuscript, the sentences often are without punctuation and illogical. I propose to focus on strengthening the review and revision in this area.
(13) Line 333-335: Don't use two "however" together.
(14) Line 344-345: “Combination of the high effective EOs with these technologies can improve shelf life of any food, including fruits like strawberries.” Any food? I think that's too absolute.
Conclusion:
(15) Line 347: “Four of 13-----“, Please write before and after the same way.
Reviewer 2 Report
The author purchased commercial EOs and applied them on postharvest preservation of strawberries, and then checked the anti B. cinerea activity of EOs. However, this was already reported before. Why it is necessary to be repeated? What is the novelty of this article? The results from different literatures are not consistent might because of the purity of the EO. If you already quantified the main compounds in the EOs, why you don't use the pure compound and to optimize a best EO? Or if you have the best EO, maybe you can combine it with the commercial fungisides to check if they can reduce the application dose and increase efficiency?
Besides, the manuscript has some language problems. For example, L49, an active ingredients...this sentence need to be revised.
The punctuation symbols should be consistent. Use capital L or small l for liter. But it MUST be consistent!! EO was used as abbreviation for essential oils at the beginning of the article. So use EO in the following sections to be consistent!!!
Reviewer 3 Report
The manuscript entitled “Antifungal Activities of Essential Oils in Vapor Phase against Botrytis cinerea and Their Potential to Control Postharvest Strawberry Gray Mold” by Dana Tančinová and co-authors provides some interesting data.
Minor remarks:
- Line 13: omit “Tel.: (optional ….
- Line 16 and throughout the test: it should be made clear to the readers that it is about strawberry fruit (berries) and not about the whole strawberry plant.
- Line 26: lemongrass and litsea EOs scored better than thyme and peppermint ones, …
Throughout the manuscript: the authors must find a proper balance as for their findings. One side is that they revealed novel ways to thwart B. cinerea by some EOs , but other (darker) side is that those ways are associated with decreased qualities in the sensory evaluation. So, all pros and all cons should be highlighted.
Lines 74-77: this sentence should be constructed as the hypothesis that was verified by conducted measurements. So, “In the present study it was hypothesized that …
Statistical analyses: why did you use two different post-hoc tests (Tukey and Duncan), what was behind that decision. Please answer.
Table 2. KMi284, 7th column: 58.2 (eucalyptus) has got similar superscript (is not different) as 0.00 (cardamom); why?. The standard deviation for eucalyptus was so high? Explain.
Reviewer 4 Report
Introduction -> Any more recent investigations of EO in vapor phase?
L152-153 Any control of applied EO amount/volume on paper?
Table 3 -> Are these numbers of samples which found growth? The detail should be clearly stated in caption or Method section.
Discussion -> Some part of discussion is difficult to follow whether it is the present results or previous investigations. More scientific discussion should be added e.g. why, how these EO gave the best/worst sensory results, mechanism of antimicrobial, why they gave different results?
L313 “did not interfere” -> because of compatible aroma or low threshold or etc???
L333 How damage? Why?
L343Add more discussion on other potential applications of these data e.g., these EO are natural compounds that can potentially be applied in active packaging to extend the shelf-life of fruits and can be a healthier choice for consumers than use of inorganic substances in active packaging (doi.org/10.1016/j.fpsl.2022.100901).
Round 2
Reviewer 2 Report
The manuscript is well-prepared, although the inovation is very low.
Reviewer 4 Report
The manuscript has been improved.